# Can we accurately forecast non-elective bed occupancy and admissions in the NHS? A time-series MSARIMA analysis of longitudinal data from an NHS Trust

Emily Eyles [1,2] Maria Theresa Redaniel,[1,2] Tim Jones [1,2] Marion Prat,[3] Tim Keen[4]

[1]The National Institute for Health Research Applied Research Collaboration West (NIHR ARC West) at University Hospitals Bristol and Weston NHS Foundation Trust, Bristol, UK
[2]Population Health Sciences, Bristol Medical School, University of Bristol, Bristol, UK
[3]School of Economics, Faculty of Social Sciences and Law, University of Bristol, Bristol, UK
[4]North Bristol NHS Trust, Westbury on Trym, UK

**Correspondence to**
Dr Emily Eyles;
emily.eyles@bristol.ac.uk

## ABSTRACT

**Objectives** The main objective of the study was to develop more accurate and precise short-term forecasting models for admissions and bed occupancy for an NHS Trust located in Bristol, England. Subforecasts for the medical and surgical specialties, and for different lengths of stay were realised

**Design** Autoregressive integrated moving average models were specified on a training dataset of daily count data, then tested on a 6-week forecast horizon. Explanatory variables were included in the models: day of the week, holiday days, lagged temperature and precipitation.

**Setting** A secondary care hospital in an NHS Trust in South West England.

**Participants** Hospital admissions between September 2016 and March 2020, comprising 1291 days.

**Primary and secondary outcome measures** The accuracy of the forecasts was assessed through standard measures, as well as compared with the actual data using accuracy thresholds of 10% and 20% of the mean number of admissions or occupied beds.

**Results** The overall Autoregressive Integrated Moving Average (ARIMA) admissions forecast was compared with the Trust's forecast, and found to be more accurate, namely, being closer to the actual value 95.6% of the time. Furthermore, it was more precise than the Trust's. The subforecasts, as well as those for bed occupancy, tended to be less accurate compared with the overall forecasts. All of the explanatory variables improved the forecasts.

**Conclusions** ARIMA models can forecast non-elective admissions in an NHS Trust accurately on a 6-week horizon, which is an improvement on the current predictive modelling in the Trust. These models can be readily applied to other contexts, improving patient flow.

## Strengths and limitations of this study

► The use of Autoregressive Integrated Moving Average models, which are simple to set up using *R*, and can be used to improve short term forecasts given enough input data are a strength of this study. Limitations include poorer accuracy in subforecasts, possibly due to using a simpler method.

► The coronavirus pandemic has an impact on the accuracy of forecasts, as shown by the sensitivity analysis.

► Data from only one NHS Trust were used, so the models are not validated for other contexts.

of emergency admissions is 3.2% on average over the last ten years.[1] Inaccurate estimates of unplanned admissions, and therefore, unpredictable emergency admissions, can lead to the cancellation of planned or routine operations.[2] This leads to subsequent need for more capacity to cope with routine operation backlog.[3 4] Granular models with daily and weekly forecasts predicted accurately can help strategically plan short-term and long-term resource management, particularly in order to cope with surges in demand.

In the past three and a half years, emergency admissions in the North Bristol NHS Trust (NBT) increased by 26% while average length of stay was reduced from 7.4 days to 6.1 days over the same period.[5] The North Bristol Trust is situated in Bristol and South Gloucestershire, England. The non-elective activity in NBT, particularly emergency admissions and bed occupancy, is higher than in other hospitals in the Bristol, North Somerset, South Gloucestershire Clinical Commissioning Group area. The consequences of this increase in emergency activity are a difficulty in meeting planned care, and, for the Trust, receiving less pay than expected, as unplanned procedures are paid less than

## INTRODUCTION

Hospitals are increasingly busy, and there is more demand for resources such as hospital beds than can be easily met. Resource management is a challenge for many hospitals, and with an increase in the number of unplanned admissions, the issue of efficient resource allocation has become more urgent. The annual growth rate in England

planned ones.[5] As the increase in demand is rising faster than demographic growth, the need to efficiently forecast non-elective activity and understand its predictors has been identified. If the Trust is better able to forecast emergency admissions and understand the factors driving their increase, then it will be able to confidently plan over the longer term. Understanding demand allows the Trust to plan what capacity it will need to meet that demand, including the beds required, the workforce and the theatre time. This in turn influences what capacity is then available for elective work.

The NBT used two main methods to forecast non-elective admissions: a scenario-based model, combining the previous year's growth in demand and demographic growth, and a crude average growth model, which only includes demographic growth.[5] These techniques have been found to be insufficient for fine-grained, accurate operational planning, especially at the specialty level.[5] Several techniques have been used to predict unplanned admissions, including multiple linear regression,[6–8] generalised estimating equations[9], exponential smoothing,[8 10] and the widely-used family of Autoregressive Integrated Moving Average (ARIMA) models.[8 11] Other work has used hybrid models, including forecasting-simulation-optimisation SARIMA and ARIMA models have previously been used to forecast emergency department admissions and occupancy,[2 8 10] emergency department crowding (hourly forecasting),[12] and infectious disease bed occupancy.[13]

Previous work has mostly used horizons of up to 1–2 weeks, as forecasting benefit was seen to be limited for longer horizons.[2] Reliable hourly models have been established,[12] but the daily horizon over a period of several weeks has been largely underexplored. Shah *et al*[14] developed a national model of daily midnight bed occupancy using 121 NHS Trusts, across a 6-week horizon, but this approach may not reflect local context, a priority in the NHS Long Term Plan.[15] Our study will explore the 6-week horizon in a site-specific context. Further, most previous research did not divide by length of stay and specialty, but rather by urgency, or how severe a case is.[10 16] Medicine and surgery may have different drivers in terms of admission to hospital, and longer lengths of stay tend to have different characteristics. Other work[17 18] has modelled not only beds but nurse and physician availability in A&E, inpatient and outpatient services.

In this paper, we aim to present a method of forecasting admissions and bed occupancy in a local context, using a 45 day (6-week) horizon. We also present subforecasts, divided by length of stay and specialty. The daily horizon over several weeks and the separation by length of stay and specialty are unique to this study, and we further use climatic information to inform the forecast as well. The subforecasts will be assessed using several measures of accuracy, including the mean absolute percentage error, and accuracy thresholds determined by the mean of that particular subforecast. This will demonstrate the possibility of conducting more detailed, specific forecasts, which will allow for better, more granular planning.

## METHODS

NBT provided operational data on non-elective, that is, unplanned, admissions to Southmead Hospital, the largest of the three hospitals in the Trust, from September 2016 to March 2020. These data are the precursor to what NBT provides to NHS Digital as part of the Hospital Episode Statistics (HES). On average, NBT reported having 868 general and acute beds available.[19] Figure 1 shows the ratio of monthly elective to non-elective admissions up

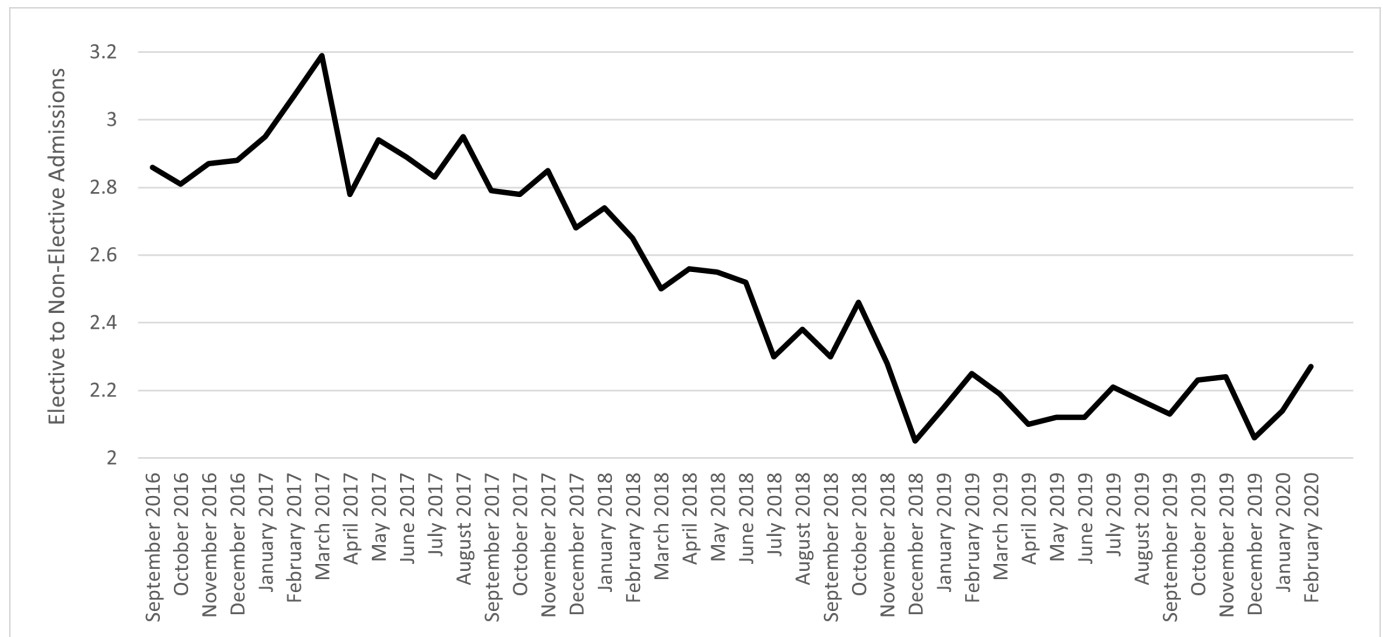

**Figure 1** Monthly ratio of elective to non-elective admissions, September 2016 to February 2020.

| Table 1 | Outcome variables and explanatory variables |
| --- | --- |
| **Variable** | **Definition** |
| Outcomes | |
| Admissions | Count of admissions to hospital, for a non-zero length of stay, that is, overnight stays |
| Bed occupancy | Count of hospital beds occupied, calculated as a count between admission and discharge dates, for non-zero lengths of stay |
| Explanatory variables | |
| Day of the week | Sunday, Monday, Tuesday, Wednesday, Thursday, Friday, Saturday |
| Temperature last year | The average temperature in Celsius on the same day last year |
| Precipitation last year | The precipitation in millimetres on the same day last year |
| Holiday | Whether a day was a bank holiday, over Christmas week, or during the Easter holiday weekend |

to February 2020, which was generated from HES data. The data from HES was obtained under licence (DARS-NIC-17875-X7K1V) from NHS Digital (previously the Health and Social Care Information Centre); Copyright 2021, reused with the permission of The Health & Social Care Information Centre. All rights reserved. The data are provided by patients and collected by the NHS as part of their care and support. HES data can be accessed via NHS Digital.[20]

The NBT data were split into two: a training dataset of 1246 days to develop the model, and a test dataset of 45 days to test the model. The 6-week forecast horizon was from the 29 January 2020 to the 14 March 2020. All days had at least one admission or occupied bed, so there were no missing values. The forecast horizon was decided by operational demand. Temperature data were obtained from the Horfield and Filton weather station, and precipitation data were obtained from the Met Office daily values for South West England.[21]

We made forecasts for admissions and bed occupancy (see table 1). Admissions are the start of a hospital spell.[22] Admissions therefore are a count of the number of patients admitted to hospital on a particular day. Bed occupancy was calculated from the admissions data by counting a bed as being occupied by a patient between their admission and discharge dates. Bed occupancy is thus a cumulative measure of how many patients are in hospital on a particular day, even if they were admitted on a previous day. Zero length of stay is defined operationally as a stay of under 24 hours, which does not overlap midnight. Patients who stay over midnight, but for under 24 hours are counted as a non-zero length of stay: for example, if the patient was admitted at 21:00 and was discharged at 03:00 the next day, this would be recorded as a length of stay of one rather than zero. Zero length of stay patients, that is, patients who did not stay over at least one midnight, were excluded from the analysis.[2] This is because zero length of stay patients are managed through a separate bed-base at NBT. The acuity of these patients means they are rarely admitted into the main bed-base and may not occupy a physical bed (eg, they may be under observation in a chair for a few hours).

We controlled for temperature and precipitation (both lagged, the same day last year) the day of the week, holidays (bank, Christmas week, and the Easter holiday weekend). The temperature and precipitation were lagged to the previous year, as weather prediction is not sufficiently accurate for longer forecast horizons. Weather data have been included in other forecasting models and contexts, to mixed results.[10 16 23 24]

Day of the week, as well as holidays, inclusive of bank holidays, were found across several studies to be highly relevant in predicting secondary care admissions.[23–26] Finally, a simple trend variable was generated, to account for the upward direction of admissions and bed occupancy over time, calculated as the days since the start of the dataset.[27] Table 1 details the variables included in the models. Separate subforecasts were also conducted divided by specialty (surgery or medicine) and length of stay (over or under 48 hours), detailed in table 2. The threshold of 48 hours was chosen as it is the acute medicine phase.[28] In the UK, as a part of good clinical care leading to successful outcomes, many patients can and should be diagnosed, treated, and even discharged within the 48-hour timeline.[28] Specialty was determined by a classification of Treatment Function Codes[29] provided by the Trust.

We used Multivariate Seasonal AutoRegressive Integrated Moving Average (MSARIMA) models to forecast emergency admissions and bed occupancy, following the ARIMA forecasting procedure detailed by Jebb et al.[27] MSARIMA models allow for seasonality, or regularly repeating cycles or trends in the data, and takes into account the autoregressive order p, d is the differencing order and q is the moving average order. The second set of brackets represents the seasonal versions of these components.[9] The autoregressive part of the model regresses the forecast variable against its own past values, to the order of the number of time lags (p).[30] The moving average part of the model uses the past regression errors terms to the order of the time lag of the error (q).[30] The differencing part of the model (d) refers to how many times lagged values have been subtracted from the data.[30]

**Table 2** Model specifications

| Outcome | Treatment specialty | Length of stay |
|---|---|---|
| Admissions | All | All |
| | | ≤48 hours |
| | | >48 hours |
| | Medicine | All |
| | | ≤48 hours |
| | | >48 hours |
| | Surgery | All |
| | | ≤48 hours |
| | | >48 hours |
| Bed occupancy | All | All |
| | | ≤48 hours |
| | | >48 hours |
| | Medicine | All |
| | | ≤48 hours |
| | | >48 hours |
| | Surgery | All |
| | | ≤48 hours |
| | | >48 hours |

The modelling component consists first of an examination of the time series to check whether it is stationary, that is, whether its mean, variance and autocorrelation do not vary over time, a necessary requirement if we want to use information about the past behaviour of the series to learn something about its future values. Unlike other time series modelling techniques, the ARIMA family of models allow for non-stationary data and the inclusion of explanatory variables.[27] Augmented Dickey-Fuller tests indicated that all data series were stationary or integrated of order 0 (d=0). MSARIMA models are a special type of regression for time series, which use information from previous days to inform them, while adjusting for other factors in the model, such as the weather, or day of the week.[10] MSARIMA modelling techniques were chosen as they are one of the recommended advanced forecasting techniques by NHS England,[22] due to the complexity and seasonality of A&E settings. Further, according to NHS England,[22] these techniques provided the most consistent estimate of daily A&E patient volumes; other methods such as artificial neural network models were said to provide less accurate forecasts of these volumes.

The mean absolute percentage error (MAPE) and the root mean square error (RMSE) are presented for each model.[2 10–13 16 30] MAPE measures model fit with closer to 0% indicating better fit.[16] For the RMSE, lower values indicate better model performance, though its value is dependent on the mean value of the variable; therefore, comparisons of this metric between models with widely varying means is hindered.[11] The subforecast means are presented in table 3 along with the SD; more detailed

results can be found in online supplemental appendix 1. The model was tested against the test dataset, providing the actual out-of-sample accuracy. The model accuracy was determined by whether the residual of the forecast versus the observed was over or under a given threshold. We used two thresholds, representing moderate and strict accuracy. This was defined as 20% of the sample mean for moderate accurate and 10% of the sample mean for strict accuracy. These thresholds match the ad-hoc thresholds determined in meetings with NBT, namely a 10/20 bed threshold for all admissions, which is matched by a 9.4/18.8 bed threshold as mean percentages. Accuracy is thus represented as a percentage of accurate daily forecasts within those thresholds (see table 3). A short-term 6-week (45 day) forecast horizon was chosen, as long-term daily forecasts require different methods, as eventually the autoregressive component of the model will converge to the mean.[30]

The data were prepared using Stata V.15.1, and analysed using the *forecast* package in R. Following the data checks, the *auto.arima* function in R was used to determine the model specifications.[31] To determine the relevance of the explanatory variables, once the model specifications were determined, the models were specified with and without the explanatory variables. Models with the explanatory variable of interest were compared with null models and models containing the other explanatory variables on the basis of AIC, similar to how *auto.arima* determines model specifications.[31] Admissions and bed occupancy were modelled separately. The first two models were all admissions and all bed occupancy data. Then, subforecasts were modelled, split by specialty or length of stay, and finally specialty and length of stay (see table 2). The code for the models is provided in online supplemental appendix 2.

Sensitivity analyses were performed for two other time periods on the overall admissions dataset. The first was using the data from September 2016 to mid December 2019, for a horizon of 6 weeks between the 1 November 2019 and the 15 December 2019. The second sensitivity analysis used the data from September 2016 to mid June 2020, for a horizon between the 2 May 2020 and the 15 June 2020. These horizons were selected to either omit the impact of the coronavirus pandemic (the earlier horizon) or to examine the forecast within the coronavirus pandemic (the later horizon). A further sensitivity analysis was performed by excluding the climatic data from the models (see online supplemental appendix 3).

### Patient and public involvement statement

Members of the public were consulted in the development of this study. The research idea was presented to them in a workshop and suggestions and comments were incorporated in the protocol. Feedback during the workshop was positive, with participants agreeing with the research objectives and the identified need for forecasting models. The use of generic and generalisable models were highlighted in the discussion, and that the audience and end

**Table 3** Model specifications and accuracy

| Model | Outcome | (Sub)forecast | SARIMA (p,d,q) (P,D,Q) | Model accuracy (strict/moderate thresholds) | RMSE | MAPE | Mean of outcome | SD of outcome |
|---|---|---|---|---|---|---|---|---|
| 1 | Admissions | All data overall | (3,0,0) (1,0,0) | 60.0%/88.9% | 9.59 | 8.66 | 94.04 | 14.98 |
| 2 | Admissions | All medical | (3,0,2) (2,0,2) | 51.1%/84.4% | 8.98 | 9.44 | 79.14 | 19.32 |
| 3 | Admissions | All surgical | (1,0,2) (0,0,0) | 28.9%/57.8% | 4.59 | 19.86 | 20.65 | 5.31 |
| 4 | Admissions | All under 48-hour length of stay | (1,0,0) (1,0,0) | 28.9%/71.1% | 6.81 | 16.53 | 35.38 | 9.06 |
| 5 | Admissions | All over 48-hour length of stay | (3,0,1) (1,0,1) | 62.2%/88.9% | 7.22 | 10.78 | 55.14 | 10.04 |
| 6 | Admissions | Under 48-hour length of stay, medical | (4,0,0) (2,0,0) | 24.4%/48.9% | 6.45 | 18.64 | 30.59 | 10.21 |
| 7 | Admissions | Under 48-hour length of stay, surgical | (0,0,1) (0,0,0) | 11.1%/31.1% | 7.82 | 44.19 | 20.99 | 8.39 |
| 8 | Admissions | Over 48-hour length of stay, medical | (5,0,1) (2,0,0) | 53.3%/82.2% | 6.88 | 11.97 | 48.55 | 12.06 |
| 9 | Admissions | Over 48-hour length of stay, surgical | (3,0,3) (1,0,2) | 25.8%/46.7% | 3.40 | 27.45 | 11.87 | 3.74 |
| 10 | Bed occupancy | All data overall | (0,0,1) (1,0,0) | 40.0%/77.8% | 158.24 | 15.57 | 823.36 | 165.22 |
| 11 | Bed occupancy | All medical | (4,0,0) (2,0,0) | 37.8%/60.0% | 151.69 | 17.06 | 760.03 | 189.22 |
| 12 | Bed occupancy | All surgical | (0,0,1) (0,0,0) | 24.4%/37.8% | 64.50 | 35.40 | 162.62 | 67.71 |
| 13 | Bed occupancy | All under 48-hour length of stay | (3,0,0) (2,0,1) | 28.9%/66.7% | 15.72 | 16.41 | 82.3 | 21.06 |
| 14 | Bed occupancy | All over 48-hour length of stay | (0,0,1) (1,0,0) | 40.0%/75.6% | 159.55 | 17.6 | 767.41 | 177.2 |
| 15 | Bed occupancy | Under 48-hour length of stay, medical | (4,0,0) (2,0,0) | 22.2%/48.9% | 14.95 | 18.63 | 70.83 | 23.95 |
| 16 | Bed occupancy | Under 48-hour length of stay, surgical | (0,0,1) (0,0,0) | 15.6%/28.9% | 7.82 | 44.19 | 21 | 8.38 |
| 17 | Bed occupancy | Over 48-hour length of stay, medical | (2,0,1) (0,0,0) | 35.6%/57.8% | 152.7 | 19.15 | 673.24 | 181.24 |
| 18 | Bed occupancy | Over 48-hour length of stay, surgical | (0,0,1) (0,0,0) | 20.0%/35.6% | 64.95 | 44.22 | 141.63 | 67.49 |

users perspectives should be considered in the development of the model and tool.

## RESULTS

Eighteen separate ARIMA models were forecast for the variable specifications described in table 1, and the subsets described in table 2. The final ARIMA specification, accuracy, RMSE, and MAPE of the models can be seen in table 3. Day of the week, holiday days, temperature and precipitation were all found to improve the accuracy of the forecasts and subforecasts. The model AICs and the AICs of comparable models without temperature and precipitation can be found in online supplemental appendix 3. The subsets' mean and SD are also provided. Models with more training data, that is, higher sample size, were more accurate.

The overall admissions model was accurate at the moderate threshold 88.9% of the time, and at the strict threshold 60.0% of the time. Bar the surgical and under 48-hour length of stay subforecasts, 82.2% of daily predictions of admissions were within the moderate admission threshold; 51.1% were within the strict accuracy admission threshold. Models for surgical admissions performed less well, with the under 48-hour length of stay surgical admissions having accuracy of 31.1% on the moderate level, and 11.1% on the strict level. This is likely due to sample size, and high variation in daily admissions.

For bed occupancy, the models generally performed more poorly. For the moderate bed threshold, all models were at least 28.9% accurate, and for the strict bed threshold, the lowest accuracy was 15.6%. However, this, similar to admissions, was dependent on specialty and

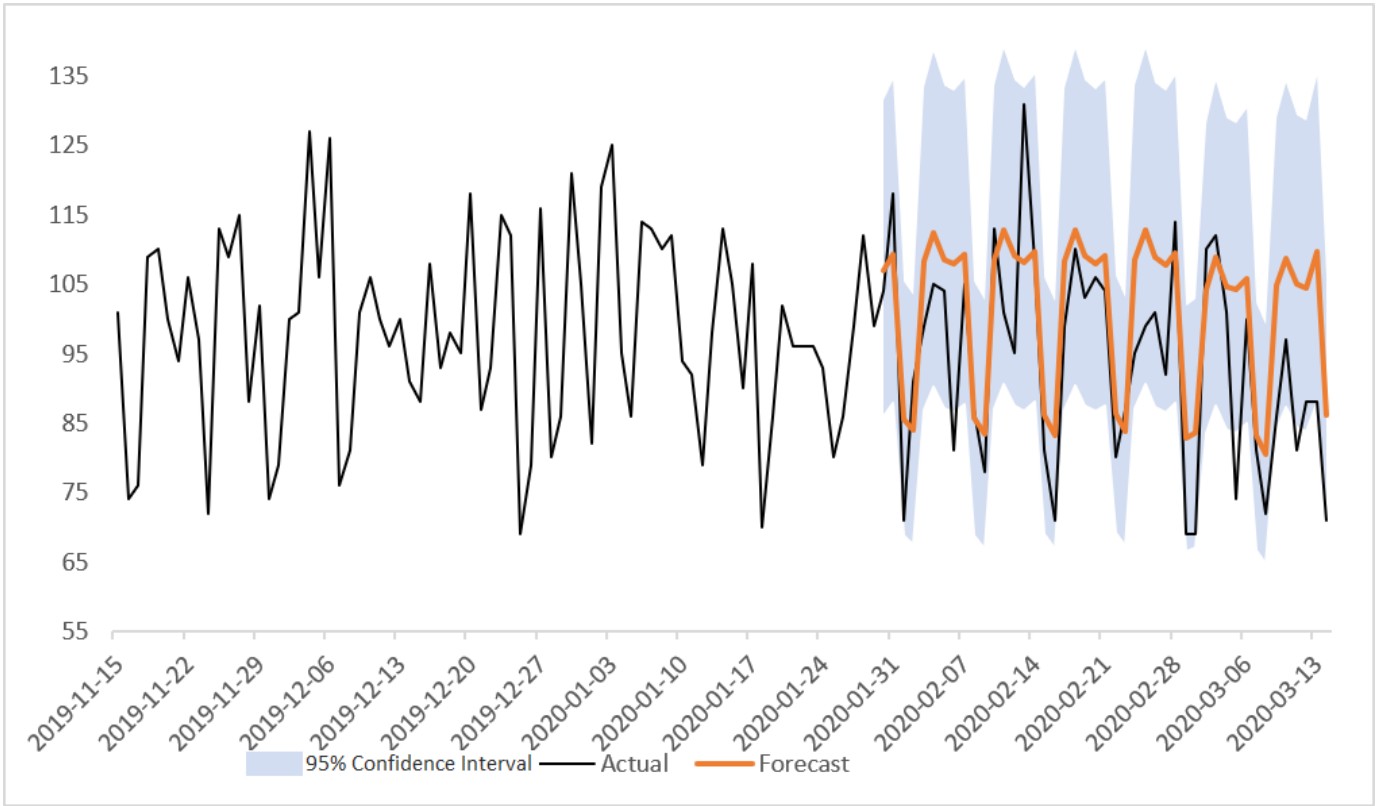

**Figure 2** Autoregressive Integrated Moving Average (ARIMA) forecast for overall admissions to North Bristol NHS Trust (NBT).

length of stay. The bed occupancy model overall (model 10) performed rather poorly, and was accurate within the moderate threshold 77.8% of the time, and within the strict threshold 40.0% of the time. Surgery performed better as a specialty, and under 48-hour length of stay tended to outperform over 48-hour length of stay as well. Online supplemental appendix 1 shows the subforecasts for each model in table 3, with 95% CIs.

Our forecasts were fairly close to the actual values observed in the test data. Our prediction for admissions overall, for example, was within the moderate threshold (18.8), excepting 5 days on our 45-day horizon. 15 days were between the moderate and strict (9.4) thresholds. 25 days were under the strict accuracy threshold. The model was less accurate the farther forward in time from the beginning of the horizon. While the forecast sometimes underestimates the number of admissions, the forecasting models are reviewed by the clinical site team, and adjustments are made to the predictor based on the particular day. Further, possibly due to the impact of the coronavirus pandemic, the latter part of the horizon may have different patterning with respect to hospital admissions. This can be seen in figure 2 (model 1 predictions), with the actual value of admissions dropping at the end of the horizon. The 95% CIs range from about 67 to 135 beds wide.

Compared with NBT's in-house prediction modelling for overall admissions, the ARIMA model outperformed them in terms of accuracy apart from 2 days, one which the NBT model was closer by just 1.5 beds, and another

where the NBT model was closer by 10 beds. The ARIMA forecast is closer in value to the true admissions count 95.6% of the time. Figure 3 shows the forecast for overall admissions and its 95% CI, compared with the actual value and NBT model. The mean difference between the ARIMA models and the actual admissions value was 7.3 beds, with a SD of 9.9 beds. The median difference for the ARIMA models was six beds. For the NBT prediction models, the mean difference from the actual value was 21 beds, with a SD of 11.1 beds, and a median difference of 21 beds. Finally, the correlation between the ARIMA prediction and the actual values was 0.75, whereas for the NBT prediction, it was −0.45. Therefore, the ARIMA models are more accurate and more precise than the NBT models.

The sensitivity analyses for the other horizons (see online supplemental appendix 4) on the overall admissions data demonstrated that the modelling strategy was more accurate prior to the coronavirus pandemic. Compared with the horizon presented earlier, the December 2019 horizon predicted within a strict accuracy threshold (10% of the sample mean) 75.6% of the time compared with 60.0% for the study horizon, and within the moderate accuracy threshold (20% of the sample mean) 93.3% of the time, compared with 88.9% in the study horizon. The June 2020 forecast horizon performed worse than both study horizon and other sensitivity analysis. It only forecasted 42.2% of the time within the strict threshold, and 77.8% of the time within the moderate threshold. The coronavirus pandemic likely has an impact on forecasting

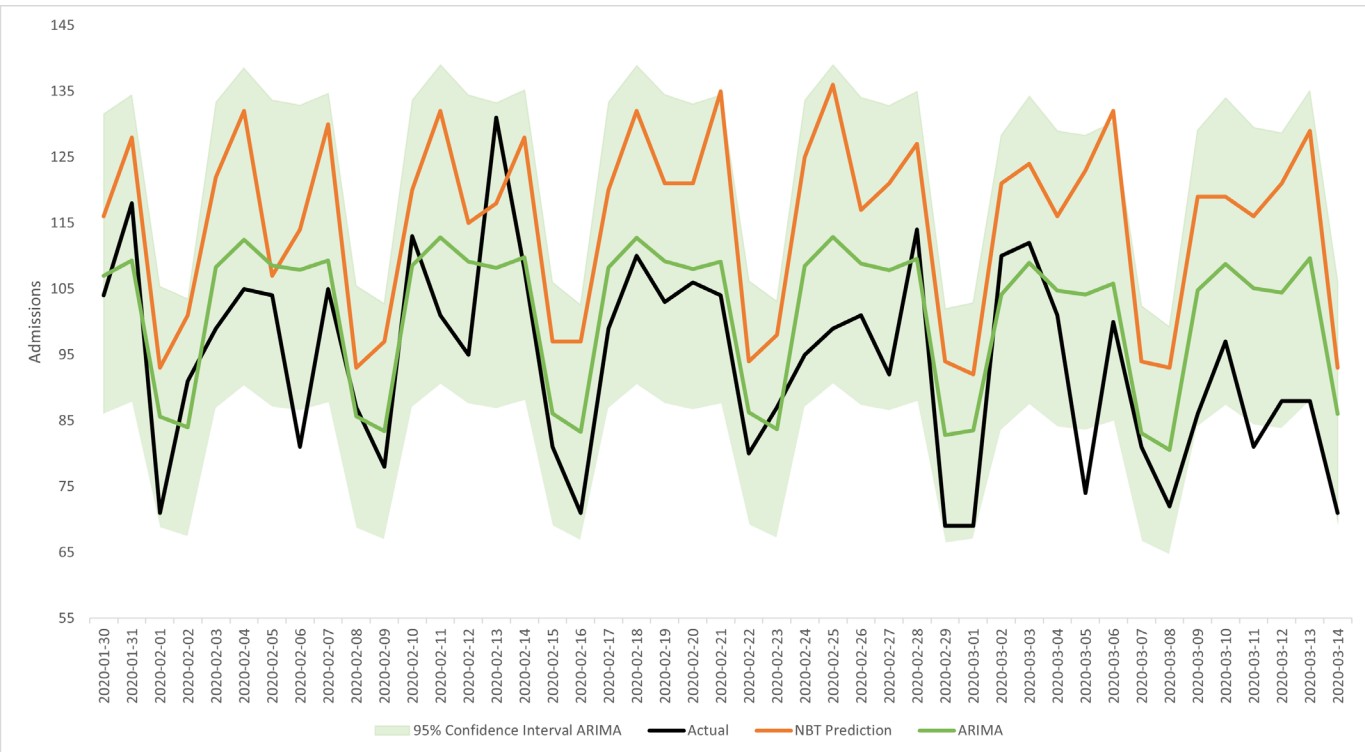

**Figure 3** Forecast predictions for overall admissions (model 1) compared with the actual values and the North Bristol NHS Trust (NBT) model. ARIMA, Autoregressive Integrated Moving Average.

models, so consideration should be taken when using data from the pandemic period.

## DISCUSSION

We have extended the literature by producing forecasts for a variety of combinations of specialty and length of stay, for both emergency admissions and bed occupancy, as well as for nonelective admissions beyond those for trauma.[23 25 32] We have found that these models work very well for some but not as well for other types of subsets, specifically shorter lengths of stay and for surgery. The models appear to work better in general for admissions rather than bed occupancy, as well as for shorter lengths of stay. The differences in accuracy can possibly be explained by the beginning of the coronavirus pandemic in the UK, as can be seen in the test data for several of the subsets, particularly for over 48 hour length of stay, and bed occupancy: a sharp drop can be seen at the beginning of February 2020. In the all data categories, and under 48-hour length of stay this sharp rise and fall can be seen around mid February 2020 (see online supplemental appendix 1). This can be interpreted as a limitation to the models' usefulness while COVID-19 has an impact on hospital admissions. Indeed, the sensitivity analyses showed that the pandemic had an impact on the accuracy of the models. Back to recovery, similarly to the pre-pandemic sensitivity analysis, the model might be more applicable although it might need more refinements to take into account changes brought about by the pandemic.

Other limitations of this study include the poorer accuracy in subforecasts, which may require more complex methods to mitigate, and the use of only one NHS Trust context. Using only one Trust means that the models are not necessarily validated for different contexts and settings. Work by Shah *et al*[14] has done similar modelling techniques at the national level, with accurate results. However, models should be site-specific to maximise utility: this means that it is necessary to test them in other Trusts. Further, Ordu *et al*[8] emphasise the importance of testing different modelling techniques for varying specialties, services and patients. More complex modelling strategies could prove to be more accurate, like the combination of different modelling techniques, such as hybrid forecasting–simulation–optimisation models,[17 18] which can be used for not only A&E but inpatient and outpatient services as well.

The strengths of this study are the use of ARIMA models, which are simple to set up using *R*, and can be used to improve short-term forecasts given enough input data. The dataset used is the hospital level precursor to the HES, available through *NHS Digital*. It is possible to improve short-term models from NBT with a relatively simple procedure, which should be transferable to other contexts, due to this similarity. The models can be incorporated easily into practice by sharing the *R* code on Github or a similar platform, and tested on data from other Trusts, which should follow a standard structure, again due to the reporting requirements of HES. The code used to generate the models in this paper is

available, with commentary, as a supplementary file (see online supplemental appendix 2).

Abraham and colleagues[2] found that admissions for patients spending at least one night were predominantly random and difficult to predict, using 3 years worth of training data. However, we found that, in general, over all subsets admissions forecasts specified better than bed occupancy ones, and that forecast horizons of over a week can be produced. It would appear that bed occupancy data are more volatile than admissions. Bed occupancy may be harder to forecast, in that it is driven by admissions, and then length of stay is at times difficult to predict.[33] It is likely that, for subsets of bed occupancy, due to the smaller amount of observations (eg, under 48 hour surgical bed occupancy has a mean of 21.00 compared with 849.71 on the whole), and higher variability of the subsets, that the models for bed occupancy produce less accurate forecasts with higher MAPEs. The MAPE for daily admissions found by Boyle *et al* is 11%: several of the subforecasts, such as the medicine, surgery, and over 48-hour length of stay ones reach this threshold, but as the amount of observations decreases, the inaccuracy seems to increase. A possible avenue of improvement would be to forecast bed occupancy by levels of severity.

Day of the week has been shown to have reasonably strong predictive capacity in previous work,[23 24 26] though the inclusion of climate variables is debated, with no clear answer.[9 10 16 24–26 32 34] Calegari *et al* found that SARIMA models including day of the week, but not climatic variables were the most accurate, especially in complex departments. However, the test period did not comprise the whole year, and the study was located in one hospital in southern Brazil, a different climate to Bristol, England.[10] Similarly, in Brazil, the ambient temperature effect was not found to improve daily visit forecasts, though again, the data only were for one season. Sun *et al*[16] also found that admissions were not predicted by weather, though caution that Singapore, the study setting, has little variation in its weather throughout the year. Atherton *et al*[25] found that adult trauma admissions were not influenced by the weather, but only used temperature in 5°C increments over a single year, whereas this study uses temperature in Celsius to a single decimal point over a longer time period. In contrast, Macgregor[34] found an association between daily temperature and trauma admissions in Aberdeen, and Rising *et al*[32] found similar results for temperature and precipitation on trauma admissions in an American hospital. Note that these solely examine trauma admissions, rather than other types of non-elective secondary admissions, which our study includes. Finally, Sahu *et al*[26] developed a Bayesian model for forecasting hospital admissions in Cardiff and Southampton, two nearby localities to Bristol, the setting of this study, and emphasised the importance and relevance of including weather data. Therefore, context is important, as in climatic data marginally improved our models, similar to other UK studies, and we suggest its inclusion be considered in future work.

By improving short-term forecasts, care can be better planned, limiting cancellations due to capacity or patient flow issues. Better care can be provided as hospitals will be less crowded. In future, other methods more appropriate for longer horizons can be tested, and medium-to-long term forecasts can be undertaken with more appropriate methods. We have developed ARIMA models that can forecast emergency admissions over a 6-week horizon with very good accuracy, which are more precise and more accurate than the current models used by the NBT. More work is needed to inform further model development, as the accuracy of subforecasts for length of stay and specialty varies significantly. This is important work because improved modelling capability will have a direct impact on the business planning of acute NHS Trusts. Improving patient flow through the hospitals greatly improves the working and care environment, reduces A&E crowding and enables hospitals to provide the most appropriate care for patients.

**Contributors** EE drafted the manuscript. EE and TJ analysed the data. MTR and TK supervised the project. TJ managed the data. MP conceptualised the methodology and conducted preliminary analysis of the data. MTR is the guarantor of this paper. The public contributed through the patient and public involvement process in the development of the study.

**Funding** This study was funded by the HDRUK Better Care Partnership (#6.12). This research was supported by the National Institute for Health Research Applied Research Collaboration West.

**Disclaimer** The views expressed in this article are those of the author(s) and not necessarily those of the NIHR or the Department of Health and Social Care. The authors are grateful to the members of the public who contributed through the patient and public involvement process.

**Competing interests** None declared.

**Patient and public involvement** Patients and/or the public were involved in the design, or conduct, or reporting, or dissemination plans of this research. Refer to the Methods section for further details.

**Patient consent for publication** Not applicable.

**Provenance and peer review** Not commissioned; externally peer reviewed.

**Data availability statement** Data may be obtained from a third party and are not publicly available. The data used in the study are collected by the NBT as part of their care and support. Sharing of anonymised data with the University of Bristol was underpinned by a data sharing agreement and solely covers the purposes of this study. Data requests can be made through the NBT.

**ORCID iDs**
Emily Eyles http://orcid.org/0000-0002-2695-7172
Tim Jones http://orcid.org/0000-0002-1199-8668

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
