## [Reviewer comments · BMJ Open]

ARTICLE DETAILS

TITLE (PROVISIONAL)	Can we accurately forecast non-elective bed occupancy and admissions in the NHS? A time-series MSARIMA analysis of longitudinal data from an NHS Trust.
AUTHORS	Eyles, Emily; Redaniel, Maria Theresa; Jones, Tim; Prat, Marion; Keen, Tim

VERSION 1 – REVIEW

REVIEWER	Boncea, Emanuela Imperial College London, Primary Care and Public Health
REVIEW RETURNED	13-Oct-2021

GENERAL COMMENTS	Overview: The authors have developed a model for forecasting non-elective admissions and bed occupancy in an NHS trust. The authors include exogenous variables such as national holidays and meteorological information from the previous year inform the model, the latter of which has not typically been widely used in a UK setting. The authors further present forecasts stratified by speciality and short/long length of stay, which is novel to their study. While the model does not perform within a set accuracy threshold in all cases, it is an improvement on in-house methods currently used by the Trust. Abstract: 1) The result that the forecast is closer in value 95.6% of the time is not stated anywhere else in the manuscript. Introduction: Major Comments: 1) Page 4, line 39: I believe at least one other study has explored forecasting nonelective hospital beds with a horizon period of 6 weeks in an NHS setting: Shah K, Sharma A, Moulton C, Swift S, Mann C, Jones S Forecasting the Requirement for Nonelective Hospital Beds in the National Health Service of the United Kingdom: Model Development Study JMIR Med Inform 2021;9(9):e21990 doi: 10.2196/21990PMID: 34591020 Minor Comments: 1) Page 3, line 33: It is a little unclear what the “previous year’s growth’ is referring to in addition to demographic growth (e.g., growth in capacity, workforce?) Methods:
--

Major Comments:

1) Page 4, lines 25-33:

- I am struggling to follow the definition of 'admissions'. I expected admissions to refer to all patients, regardless of whether they occupied a bed or not. However, as I understand, all patients which have not occupied an overnight bed ('zero length of stay') are excluded. Therefore, if admissions necessarily occupy a bed, would you clarify the difference between the populations included in 'admissions' and those included in 'bed occupancy'?
- If indeed all patients with a LOS <24 hours have been removed from the analysis, would you consider that their exclusion is a limitation of your study? Short LOS admissions make up a considerable proportion of patients using acute medical units which are not accounted for
- If all patients with LOS <24 hours were removed, consider amending the relevant sections of Table 2 to ≥ 24 hours & ≤ 48 hours for clarity

2) It would be helpful to add an overview of the number of acute beds available in the trust so that the reader has an upper limit of bed space for context, and an estimate of the proportion of elective and non-elective admissions.

3) Could you briefly explain the rationale behind splitting length of stay at (≥ 24 hours &) ≤ 48 hours and >48 hours? As it resulted in small sample sizes, why not >72 hours?

4) How were medicine and surgical specialities defined? Were they stratified by Treatment Function Codes, or were OPCS-4 codes used?

5) It would be helpful to see the difference in AICs in your model building process (Page 7, lines 6-9) at least in an appendix, as the fact that the model was improved by climatic data is an interesting finding

Minor Comments:

- 1) Page 4, lines 25-33: I would suggest beginning with a definition of 'zero' length of stay followed by the explanation of why you excluded it for clarity
- 2) Page 4, line 39-50: Consider whether some of the commentary around previous studies using weather data would fit better within the introduction or discussion rather than methods section
- 3) The link to reference 19 is not working

Results:

Major Comments:

1) Page 9, line 35: It is stated that the Covid pandemic may have impacted the accuracy of your results (and this is reiterated in the discussion); could you provide a sensitivity analysis forecasting only up to December 2019, as this should omit any impact from the pandemic?

2) Table 4 (page 10):

- While I think the information presented is really useful and transparent, it is difficult to distinguish the shades of grey on the screen. I would suggest visualising table 4 as a line graph, e.g. with dates on the x-axis, and admissions on the y axis.
- On several instances the SARIMA prediction underestimates the number of admissions, often failing to pick up peaks in

	admissions/beds occupied. Depending on the hospital's acute bed capacity, underestimating admissions may be more of a concern for patient safety. Could you comment on whether this would be an issue if it were to be implemented in the trust?  • I am struggling to understand why some models which have good accuracy in terms of the set thresholds have higher MAPE values than others with poorer accuracy (e.g., model 10 has a moderate accuracy of 77.8 and a MAPE of 158.24 versus model 16 which has a moderate accuracy of 28.9 but an MAPE of 7.82) Would you be able to explain? Discussion: Major Comments: 1) It is stated that the model may be less accurate considering the pandemic; I think this needs to be rephrased as a limitation of the model's usefulness while Covid-19 still has an impact on hospital admissions. Refining the model to include e.g., infection rate and lockdown information could also be part of future work. Appendix: 1) Please add a y-axis label to the graphs, and define the blue shading. 2) I would double check the page numbers of the TRIPOD checklist. I believe limitations should be on page 12, and there are no implications on page 14 so it may be that all pages should be - 1.
--	--

REVIEWER	Ordu, Muhammed Osmaniye Korkut Ata Universitesi, Department of Industrial Engineering
REVIEW RETURNED	14-Oct-2021

GENERAL COMMENTS	In the study, the authors carried out studies about the bed occupancy rates and demand forecasts of an NHS hospital. In the study, the demands of specialities and total demands were forecasted. ARIMA, which is one of traditional time series technique, was used as a method. Tests were conducted over a 6-week horizon. Four different independent variables were used in the study. Forecasting can always have a contribution to the knowledge with new and up-to-date data. In particular, the disruptions in health systems experienced during the Covid19 pandemic have revealed the importance of knowing or predicting hospital demands in advance. It shows that it is of vital importance in terms of providing information about whether the hospital's resources will be sufficient, especially with non-elective bed occupancy rates. The article can be published after the following issues are resolved. 1. The literature part of the article is both insufficient and out of date. Below are some very recent publications. These publications have forecasted patient demand of one of the UK's NHS hospitals. In addition, the number of beds and human resources needed has been optimized. - A Hybrid Analytical Model for an Entire Hospital Resource Optimisation. Soft Computing, (2021), 25(17), 11673-11690. https://doi.org/10.1007/s00500-021-06072-x - A novel healthcare resource allocation decision support tool: A forecasting-simulation-optimization approach. Journal of the
--

	Operational Research Society, 72(3), 485-500. https://doi.org/10.1080/01605682.2019.1700186 - A comprehensive modelling framework to forecast the demand for all hospital services. The International Journal of Health Planning and Management, 34(2), e1257–e1271. http://dx.doi.org/10.1002/hpm.2771 2. Does the Covid19 pandemic have an effect on the data used in the study? 3. Why was only the ARIMA method chosen, although there are other successful methods in the literature? Please convince the readers and reviewers? 4. In literature and in the articles given in question 2 above, the days before and after the holiday are used as an independent variable. Why were these variables not taken into account in this study? 5. Why is 48 hours considered as a threshold in Table 2? Please explain.
--	---

VERSION 1 – AUTHOR RESPONSE

Reviewer 1 comment	Letter reply	
Abstract: 1) The result that the forecast is closer in value 95.6% of the time is not stated anywhere else in the manuscript.	This result was missed from the Results section in error. It was in the (now removed, and made into a figure) Table 4. The text in the Results section has been amended.	‘The ARIMA forecast is closer in value to the true admissions count 95.6% of the time. Figure 3 shows the forecast for overall admissions and its 95% confidence interval, compared to the actual value and NBT model.’ P10
Introduction: Major Comments: 1) Page 4, line 39: I believe at least one other study has explored forecasting nonelective hospital beds with a horizon period of 6 weeks in an NHS setting: Shah K, Sharma A, Moulton C, Swift S, Mann C, Jones S Forecasting the Requirement for Nonelective Hospital Beds in the National Health Service of the United Kingdom: Model Development Study JMIR Med Inform 2021;9(9):e21990 doi: 10.2196/21990 PMID: 34591020	We acknowledge the study by Shah et al, and appreciate the reviewer flagging it up for us. As we submitted this manuscript prior to publication of this study, it was not included in our literature search, but we have amended the text accordingly to acknowledge the	‘Shah et al14 developed a national model of daily midnight bed occupancy using 121 NHS Trusts, across a six week horizon, but this approach may not reflect local context, a priority in the NHS Long Term Plan15. Our study will explore the six week horizon in a site-specific context.’ P3 ‘Work by Shah et al14 has done similar modelling techniques at the national

	work of Shah et al, and revised the text about our horizon period.	level, with accurate results. However, models should be site-specific to maximise utility: this means that it is necessary to test them in other Trusts.' P11
Minor Comments: 1) Page 3, line 33: It is a little unclear what the "previous year's growth" is referring to in addition to demographic growth (e.g., growth in capacity, workforce?)	The text has been revised to specify growth in demand.	'The NBT used two main methods to forecast non-elective admissions: a scenario-based model, combining the previous year's growth in demand and demographic growth, and a crude average growth model, which only includes demographic growth5" p3
Methods: Major Comments: 1) Page 4, lines 25-33: • I am struggling to follow the definition of 'admissions'. I expected admissions to refer to all patients, regardless of whether they occupied a bed or not. However, as I understand, all patients which have not occupied an overnight bed ('zero length of stay') are excluded. Therefore, if admissions necessarily occupy a bed, would you clarify the difference between the populations included in 'admissions' and those included in 'bed occupancy'?	The text has been clarified to define and differentiate admissions and bed occupancy.	'Admissions therefore are a count of the number of patients admitted to hospital on a particular day. Bed occupancy was calculated from the admissions data by counting a bed as being occupied by a patient between their admission and discharge dates. Bed occupancy is thus a cumulative measure of how many patients are in hospital on a particular day, even if they were admitted on a previous day.' P4
If indeed all patients with a LOS <24 hours have been removed from the analysis, would you consider that their exclusion is a limitation of your study? Short LOS admissions make up a considerable proportion of patients using acute medical units which are not accounted for	The methods section has been amended clarify zero length of stay and why it was excluded. Specifically, it is because zero LOS patients are managed through a separate bed-base, and are rarely admitted into the main bed base (e.g. under observation in a chair, rather than in a physical bed).	'Zero length of stay is defined operationally as a stay of under 24 hours, which does not overlap midnight. Patients who stay over midnight, but for under 24 hours are counted as a non zero length of stay: e.g. if the patient was admitted at 9pm and was discharged at 3am the next day, this would be recorded as a length of stay of one rather than zero. Zero length of stay patients, i.e. patients who did not stay over at least one midnight, were excluded from the analysis2. This is because zero length of stay patients are managed through a separate bed-base at NBT.

		The acuity of these patients means they are rarely admitted into the main bed-base and may not occupy a physical bed (e.g., they may be under observation in a chair for a few hours).’ P4
If all patients with LOS <24 hours were removed, consider amending the relevant sections of Table 2 to >=24 hours & <= 48 hours for clarity	A length of stay of greater than zero can include patients who stayed less than 24 hours, and this has been clarified in text, so we have not amended the thresholds.	‘Patients who stay over midnight, but for under 24 hours are counted as a non zero length of stay: e.g. if the patient was admitted at 9pm and was discharged at 3am the next day, this would be recorded as a length of stay of one rather than zero’ p4
2) It would be helpful to add an overview of the number of acute beds available in the trust so that the reader has an upper limit of bed space for context, and an estimate of the proportion of elective and non-elective admissions.	We have provided data from the NHS reporting the average number of general and acute beds available in NBT. We have provided a figure of the proportion of elective to non-elective admissions for the months in the dataset.	‘On average, NBT reported having 868 General and Acute beds available ¹⁹ . Figure 1 shows the ratio of monthly elective to non-elective admissions up to February 2020, which was generated from HES data.’ P4
3) Could you briefly explain the rationale behind splitting length of stay at (>=24 hours & <= 48 hours and >48 hours)? As it resulted in small sample sizes, why not >72 hours?	The rationale was operational, and dictated by the Trust, specifically the 48 hour timeline as critical in acute medicine.	‘The threshold of 48 hours was chosen as it is the acute medicine phase ²⁸ . In the UK, as a part of good clinical care leading to successful outcomes, many patients can and should be diagnosed, treated, and even discharged within the 48 hour timeline ²⁸ .’ P5
4) How were medicine and surgical specialities defined? Were they stratified by Treatment Function Codes, or were OPCS-4 codes used?	They were defined by Treatment Function Codes, with a guide by specialty provided by the Trust. The	‘Specialty was determined by a classification of Treatment Function Codes ²⁹ provided by the Trust.’ P5

	text has been clarified.	
5) It would be helpful to see the difference in AICs in your model building process (Page 7, lines 6-9) at least in an appendix, as the fact that the model was improved by climatic data is an interesting finding	An appendix table (Appendix 3) has been provided with AICs for all models both with and without climatic data.	'A further sensitivity analysis was performed by excluding the climatic data from the models (see Appendix 3).' P7 Please see Appendix 3 (supplementary files)
Minor Comments: 1) Page 4, lines 25-33: I would suggest beginning with a definition of 'zero' length of stay followed by the explanation of why you excluded it for clarity	The text has been re-ordered and slightly amended.	'Zero length of stay is defined operationally as a stay of under 24 hours, which does not overlap midnight. Patients who stay over midnight, but for under 24 hours are counted as a non zero length of stay: e.g. if the patient was admitted at 9pm and was discharged at 3am the next day, this would be recorded as a length of stay of one rather than zero. Zero length of stay patients, i.e. patients who did not stay over at least one midnight, were excluded from the analysis ² . This is because zero length of stay patients are managed through a separate bed-base at NBT. The acuity of these patients means they are rarely admitted into the main bed-base and may not occupy a physical bed (e.g., they may be under observation in a chair for a few hours).' P4
2) Page 4, line 39-50: Consider whether some of the commentary around previous studies using weather data would fit better within the introduction or discussion rather than methods section	The commentary around weather data has been moved to the discussion.	'Day of the week has been shown to have reasonably strong predictive capacity in previous work ^{23 24 26} , though the inclusion of climate variables is debated, with no clear answer ^{9 10 16 24-26 32 34} . Calegari and colleagues found that SARIMA models including day of the week, but not climatic variables were the most accurate, especially in complex departments. However, the test period did not comprise the whole year,

	and the study was located in one hospital in southern Brazil, a different climate to Bristol, England¹⁰. Similarly, in Brazil, the ambient temperature effect was not found to improve daily visit forecasts, though again, the data only were for one season. Sun et al¹⁶ also found that admissions were not predicted by weather, though caution that Singapore, the study setting, has little variation in its weather throughout the year. Atherton et al²⁵ found that adult trauma admissions were not influenced by the weather, but only used temperature in 5 Celsius increments over a single year, whereas this study uses temperature in Celsius to a single decimal point over a longer time period. In contrast, Macgregor³⁴ found an association between daily temperature and trauma admissions in Aberdeen, and Rising et al³² found similar results for temperature and precipitation on trauma admissions in an American hospital. Note that these solely examine trauma admissions, rather than other types of nonelective secondary admissions, which our study includes. Finally, Sahu et al²⁶ developed a Bayesian model for forecasting hospital admissions in Cardiff and Southampton, two nearby localities to Bristol, the setting of this study, and emphasised the importance and relevance of including weather data. Therefore context is important, as in climatic data marginally improved our
--	--

		models, similar to other UK studies, and we suggest its inclusion be considered in future work.' P12
3) The link to reference 19 is not working	The links have all been tested prior to resubmission, and should now be working.	
Results: Major Comments: 1) Page 9, line 35: It is stated that the Covid pandemic may have impacted the accuracy of your results (and this is reiterated in the discussion); could you provide a sensitivity analysis forecasting only up to December 2019, as this should omit any impact from the pandemic?	A new appendix has been added with two sensitivity analyses: one with your suggested horizon (up to December 2019), and one suggested by the other reviewer, which is up to June 2020.	'The sensitivity analyses for the other horizons (see Appendix 4) on the overall admissions data demonstrated that the modelling strategy was more accurate prior to the coronavirus pandemic. Compared to the horizon presented earlier, the December 2019 horizon predicted within a strict accuracy threshold (10% of the sample mean) 75.6% of the time compared to 60.0% for the study horizon, and within the moderate accuracy threshold (20% of the sample mean) 93.3% of the time, compared to 88.9% in the study horizon. The June 2020 forecast horizon performed worse than both study horizon and other sensitivity analysis. It only forecasted 42.2% of the time within the strict threshold, and 77.8% of the time within the moderate threshold. The coronavirus pandemic likely has an impact on forecasting models, so consideration should be taken when using data from the pandemic period.' P10
2) Table 4 (page 10):  While I think the information presented is really useful and transparent, it is difficult to distinguish the shades of grey on the screen. I would suggest visualising table 4 as a line 	We have made table 4 into a line graph, and appreciate the suggestion for	Please see figure on page 10.

graph, e.g. with dates on the x-axis, and admissions on the y axis.	readability of the paper.	
On several instances the SARIMA prediction underestimates the number of admissions, often failing to pick up peaks in admissions/beds occupied. Depending on the hospital's acute bed capacity, underestimating admissions may be more of a concern for patient safety. Could you comment on whether this would be an issue if it were to be implemented in the trust?	It is not considered an issue from a patient safety perspective. According to the Trust: "It is only a model – our clinical site team review the predictions each day and compare to what has been experienced and they will make adjustments to the predictor where it feels too high/low on a particular day. A more accurate model helps though – it should give us more confidence in the predictor."	'While the forecast sometimes underestimates the number of admissions, the forecasting models are reviewed by the clinical site team, and make adjustments to the predictor based on the particular day.' P10
I am struggling to understand why some models which have good accuracy in terms of the set thresholds have higher MAPE values than others with poorer accuracy (e.g., model 10 has a moderate accuracy if 77.8 and a MAPE of 158.24 versus model 16 which has a moderate accuracy of 28.9 but an MAPE of 7.82) Would you be able to explain?	Thank you for flagging up the issue with the table. On checking the source table for the version in the manuscript, the RMSE and MAPE columns were mislabelled and thus transposed. The column headings have been corrected, and the corresponding values checked.	The heading was fixed, please see pages 8-9 for table 3.
Discussion Major Comments: 1) It is stated that the model may be less accurate considering the pandemic; I think this needs to be rephrased as a limitation of the model's usefulness while Covid-19 still has an impact on hospital admissions. Refining the model to include e.g., infection rate and	We have added to the discussion to interpret this as a limitation, to include the result from the sensitivity analyses, and slightly reordered the text to	'This can be interpreted as a limitation to the models' usefulness while COVID-19 has an impact on hospital admissions. Indeed, the sensitivity analyses showed that the pandemic had an impact on the accuracy of the models. Back to recovery,

lockdown information could also be part of future work.	group limitations together.	similarly to the pre-pandemic sensitivity analysis, the model might be more applicable although it might need more refinements to take into account changes brought about by the pandemic.' P11
Appendix: 1) Please add a y-axis label to the graphs, and define the blue shading.	We have added y axis labels, and defined the blue shading in the legend.	Please see Appendix 1
2) I would double check the page numbers of the TRIPOD checklist. I believe limitations should be on page 12, and there are no implications on page 14 so it may be that all pages should be -1.	The TRIPOD checklist has been revised and the page numbers checked. Thank you for bringing this to our attention.	Please see revised TRIPOD checklist.
Reviewer 2		
1. The literature part of the article is both insufficient and out of date. Below are some very recent publications. These publications have forecasted patient demand of one of the UK's NHS hospitals. In addition, the number of beds and human resources needed has been optimized. - A Hybrid Analytical Model for an Entire Hospital Resource Optimisation. Soft Computing, (2021), 25(17), 11673-11690. https://doi.org/10.1007/s00500-021-06072-x - A novel healthcare resource allocation decision support tool: A forecasting-simulation-optimization approach. Journal of the Operational Research Society, 72(3), 485-500. https://doi.org/10.1080/01605682.2019.1700186 - A comprehensive modelling framework to forecast the demand for all hospital services. The International Journal of Health Planning and Management, 34(2), e1257–e1271. http://dx.doi.org/10.1002/hpm.2771	We thank you for the additional references. We have missed some of them as our paper was submitted before their publication. We have updated our literature review and discussion accordingly to account for these innovations.	'Several techniques have been used to predict unplanned admissions, including multiple linear regression⁶⁻⁸, generalised estimating equations⁹, exponential smoothing^{8 10}, and the widely-used family of Autoregressive Integrated Moving Average (ARIMA) models^{8 11}. Other work has used hybrid models, including forecasting-simulation-optimisation SARIMA and ARIMA models have previously been used to forecast emergency department admissions and occupancy^{2 8 10}, emergency department crowding (hourly forecasting)¹², and infectious disease bed occupancy¹³.' P3 'Other work^{17 18} has modelled not only beds but nurse and physician availability in A&E, inpatient and outpatient services.'

		'Further, Ordu et al⁸ emphasise the importance of testing different modelling techniques for varying specialties and services. patients. More complex modelling strategies could prove to be more accurate, like the combination of different modelling techniques, such as hybrid forecasting-simulation-optimisation models^{17 18}, which can be used for not only A&E but inpatient and outpatient services as well.' P11
2. Does the Covid19 pandemic have an effect on the data used in the study?	The pandemic appears to have had an effect on the accuracy of our forecasts. We have done two sensitivity analyses: one on data up to December 2019, and another on data up to June 2020. These sensitivity analyses have shown that the pandemic has had an impact on the data, and the text has been amended accordingly.	'Sensitivity analyses were performed for two other time periods on the overall admissions dataset. The first was using the data from September 2016 to mid December 2019, for a horizon of six weeks between the 1 November 2019 and the 15 December 2019. The second sensitivity analysis used the data from September 2016 to mid June 2020, for a horizon between the 2 May 2020 and the 15 June 2020. These horizons were selected to either omit the impact of the coronavirus pandemic (the earlier horizon) or to examine the forecast within the coronavirus pandemic (the later horizon). A further sensitivity analysis was performed by excluding the climatic data from the models (see Appendix 3).' P7 'The sensitivity analyses for the other horizons (see Appendix 4) on the overall admissions data demonstrated that the modelling strategy was more accurate prior to the coronavirus pandemic.

		Compared to the horizon presented earlier, the December 2019 horizon predicted within a strict accuracy threshold (10% of the sample mean) 75.6% of the time compared to 60.0% for the study horizon, and within the moderate accuracy threshold (20% of the sample mean) 93.3% of the time, compared to 88.9% in the study horizon. The June 2020 forecast horizon performed worse than both study horizon and other sensitivity analysis. It only forecasted 42.2% of the time within the strict threshold, and 77.8% of the time within the moderate threshold. The coronavirus pandemic likely has an impact on forecasting models, so consideration should be taken when using data from the pandemic period.' P10 'This can be interpreted as a limitation to the models' usefulness while COVID-19 has an impact on hospital admissions. Indeed, the sensitivity analyses showed that the pandemic had an impact on the accuracy of the models. Back to recovery, similarly to the pre-pandemic sensitivity analysis, the model might be more applicable although it might need more refinements to take into account changes brought about by the pandemic.'p11
3. Why was only the ARIMA method chosen, although there are other successful methods in the literature? Please convince the readers and reviewers?	We have added justification of our use of ARIMA models to the methods. We sought to use a method that was understandable by and transferable to	'Unlike other time series modelling techniques, the ARIMA family of models allow for nonstationary data and the inclusion of explanatory variables²⁷'[6 'MSARIMA modelling techniques were chosen as

	NHS Trusts. Furthermore, it is a technique highly recommended by NHS England to forecast in A&E settings (see citation 22).	they are one of the recommended advanced forecasting techniques by NHS England²², due to the complexity and seasonality of A&E settings. Further, according to NHS England²², these techniques provided the most consistent estimate of daily A&E patient volumes; other methods such as artificial neural network models were said to provide less accurate forecasts of these volumes. 'p6
4. In literature and in the articles given in question 2 above, the days before and after the holiday are used as an independent variable. Why were these variables not taken into account in this study?	The Trust did not note changes in demand on the days before and after a bank holiday, so they were not included in the analysis. Please note that Christmas week and the Easter holiday weekend were included in the analysis, and the text has been clarified to reflect this.	'We controlled for temperature and precipitation (both lagged, the same day last year) the day of the week, holidays (bank, Christmas week, and the Easter holiday weekend). The temperature and precipitation were lagged to the previous year, as weather prediction is not sufficiently accurate for longer forecast horizons' p5
5. Why is 48 hours considered as a threshold in Table 2? Please explain.	The decision to use above and below 48 hours length of stay was based in operational reasons when deciding on the analytic strategy with the Trust. The text has been amended to reflect this.	'The threshold of 48 hours was chosen as it is the acute medicine phase²⁸. In the UK, as a part of good clinical care leading to successful outcomes, many patients can and should be diagnosed, treated, and even discharged within the 48 hour timeline²⁸. Specialty was determined by a classification of Treatment Function Codes²⁹ provided by the Trust.'

VERSION 2 – REVIEW

REVIEWER	Boncea, Emanuela Imperial College London, Primary Care and Public Health
REVIEW RETURNED	07-Mar-2022

GENERAL COMMENTS	Thank you for the opportunity to review this interesting and useful paper, and thank you to the authors for all their additional work on the manuscript following the first review. The authors have addressed all of my comments, but I want to suggest a few small amendments, which are mostly semantic, that would make the paper more coherent in some sections:  • Table 3: after clarifying the error regarding MAPE/RMSE being transposed the values make much more sense. It would help if you added ‘%’ to the top of the MAPE table column to indicate the values given are percentages. Secondly, as some of the means in the models vary widely, comparing model RMSE values between these is trivial. When defining RMSE it may be helpful to explicitly say (as an example):  o “For the RMSE, lower values indicate better model performance, though its value is dependent on the mean value of the variable and therefore comparisons of this metric between models with widely varying means is hindered (11). The model means and standard deviations are also presented in table 3.” • Table 3: model 14 and 7 appear to be missing a comma in the “SARIMA” column and model 2 has an extra comma • Page 5 line 10: the sentence does not make sense – delete the word ‘to’ or the word ‘with’ • Page 10, line 12: the sentence does not make sense – change “and make adjustments to the predictor based on the particular day” to “and adjustments are made to the predictor based on the particular day.” • Page 11, line 28: the sentence does not make sense – change “varying specialties and services. patients.” to “varying specialties, services and patients” • Page 11, lines 34-42: two sentences are missing full stops • Please add the 95% CI label to the blue shading of Figure 2 also
---

REVIEWER	Ordu, Muhammed Osmaniye Korkut Ata Universitesi, Department of Industrial Engineering
REVIEW RETURNED	25-Feb-2022

GENERAL COMMENTS	The changes made by the authors are adequate. It can be accepted for publication.
---

VERSION 2 – AUTHOR RESPONSE

Reviewer comment	Response/Change to Text
Reviewer: 2 Dr. Muhammed Ordu, Osmaniye Korkut Ata Universitesi Comments to the Author: The changes made by the authors are adequate. It can be accepted for publication.	Thank you for reviewing and approving our revisions, Dr. Ordu.
Reviewer: 1 Dr. Emanuela Boncea, Imperial College London Comments to the Author: Thank you for the opportunity to review this interesting and useful paper, and thank you to the authors for all their additional work on the manuscript following the first review. The authors have addressed all of my comments, but I want to suggest a few small amendments, which are mostly semantic, that would make the paper more coherent in some sections:	Thank you Dr. Boncea for reviewing the revisions to the paper and suggesting these amendments.
 • Table 3: after clarifying the error regarding MAPE/RMSE being transposed the values make much more sense. It would help if you added '%' to the top of the MAPE table column to indicate the values given are percentages. Secondly, as some of the means in the models vary widely, comparing model RMSE values between these is trivial. When defining RMSE it may be helpful to explicitly say (as an example):  o “For the RMSE, lower values indicate better model performance, though its value is dependent on the mean value of the variable and therefore comparisons of this metric between models with widely varying means is hindered (11). The model means and standard deviations are also presented in table 3.” 	We have mostly used your suggested example text to better to be more explicit about the comparability of RMSE between models. “For the RMSE, lower values indicate better model performance, though its value is dependent on the mean value of the variable, therefore comparisons of this metric between models with widely varying means is hindered (11). The subforecast means are presented in table 3 along with the standard deviations.” (page 7)
 • Table 3: model 14 and 7 appear to be missing a comma in the “SARIMA” column and model 2 has an extra comma 	The commas have been fixed, please see page 8-9.

 Page 5 line 10: the sentence does not make sense – delete the word ‘to’ or the word ‘with’ 	The word ‘with’ has been deleted (page 5)
 Page 10, line 12: the sentence does not make sense – change “and make adjustments to the predictor based on the particular day” to “and adjustments are made to the predictor based on the particular day.” 	We have amended the text with this suggestion (Page 10)
 Page 11, line 28: the sentence does not make sense – change “varying specialties and services. patients.” to “varying specialities, services and patients” 	The text has been corrected. (page 11)
 Page 11, lines 34-42: two sentences are missing full stops 	The full stops have been corrected (page 11)
 Please add the 95% CI label to the blue shading of Figure 2 also 	The 95% CI label has been added to Figure 2.